# UniFlow-Audio: Unified Flow Matching for Audio Generation from Omni-Modalities

## Abstract

Audio generation, including speech, music and sound effects, has advanced rapidly in recent years. These tasks can be divided into two categories: time-aligned (TA) tasks, where each input unit corresponds to a specific segment of the output audio (e.g., phonemes aligned with frames in speech synthesis); and non-time-aligned (NTA) tasks, where such alignment is not available. Since modeling paradigms for the two types are typically different, research on different audio generation tasks has traditionally followed separate trajectories. However, audio is not inherently divided into such categories, making a unified model a natural and necessary goal for general audio generation. Previous unified audio generation works have adopted autoregressive architectures, while unified non-autoregressive approaches remain largely unexplored. In this work, we propose UniFlow-Audio, a universal audio generation framework based on flow matching. We propose a dual-fusion mechanism that temporally aligns audio latents with TA features and integrates NTA features via cross-attention in each model block. Task-balanced data sampling is employed to maintain strong performance across both TA and NTA tasks. UniFlow-Audio supports omni-modalities, including text, audio, and video. By leveraging the advantage of multi-task learning and the generative modeling capabilities of flow matching, UniFlow-Audio achieves strong results across 7 tasks using fewer than 8K hours of public training data and under 1B trainable parameters. Even the small variant with only $\sim$200M parameters shows competitive performance, highlighting UniFlow-Audio as a potential non-auto-regressive foundation model for audio generation. Code and models will be available at https://anonymous3387a8c.github.io/uniflow_audio.

## 1 Introduction

With the rapid evolution of generative models (Vaswani et al., 2017; Ho et al., 2020; Lipman et al., 2023), recent works have achieved remarkable improvements in generation quality (Esser et al., 2024; Polyak et al., 2024), promoting popularity of artificial intelligence generated content (AIGC). As an important modality, audio has also made remarkable progress in various generation tasks, with text-to-speech synthesis (TTS) (Wang et al., 2023a) and text-to-audio (T2A) generation (Liu et al., 2023) serving as representative tasks. Traditional audio generation models are designed for specific tasks, such as converting text to speech or music. This paradigm is suboptimal, as it overlooks the interconnected nature of real-world auditory information.

To overcome this limitation, we aim at a unified framework for audio generation that accommodates diverse input (text, audio, video) and output modalities (speech, music, sound effect). We observe that despite their differences, these tasks can be fundamentally categorized by the temporal relationship between input and output: either *time-aligned (TA)* or *non-time-aligned (NTA)*, as shown in Figure 1. For TA tasks, there is strict temporal alignment between input and output, such as the monotonic alignment in text-to-speech (TTS), the one-to-one frame alignment in speech enhancement (SE), and one-to-$N$ frame alignment in video-to-audio (V2A). In contrast, NTA tasks, such as T2A, do not require such a temporal alignment constraint: the input sequence (textual description) corresponds holistically to the entire output soundscape, with semantic consistency being the primary objective rather than temporal correspondence. This fundamental difference in alignment requirements has historically necessitated specialized modeling approaches for TA and NTA tasks.

Figure 1: Illustration of time-aligned (TA) tasks and non-time-aligned (NTA) tasks.

While recent works have explored unified audio generation with autoregressive (AR) architectures, unified non-autoregressive (NAR) approaches remain relatively underexplored. UniAudio (Yang et al., 2024) adopts an AR paradigm, achieving strong zero-shot performance on both AR and NAR tasks. However, AR models rely on sequential decoding and discrete tokenizers, whereas NAR models generate continuous audio representations in parallel, which may offer advantages in latency and quality. Moreover, AR models rely entirely on self-attention to learn input–target alignment, which can be unstable for long sequences or in low-resource settings. The inherent exposure bias issue exacerbates this challenge. In addition, the teacher-forcing training strategy used in Transformer-based AR decoders introduces the well-known exposure bias problem, causing errors to accumulate as the generated sequence becomes longer. Thus, NAR-based unified audio generation remains worth exploring. AudioX represents an NAR attempt, but it focuses exclusively on NTA tasks and cannot handle TA tasks such as TTS, which require variable-length generation. Meanwhile, task-specific NAR models like VoiceFlow (Guo et al., 2024) perform well on TA tasks by temporally aligning content embeddings with audio latents, yet this modeling paradigm does not generalize to NTA tasks. This leaves a gap for a single NAR framework capable of unifying both TA and NTA tasks within one modeling paradigm.

In this work, we propose UniFlow-Audio, a universal audio generation framework based on flow matching that unifies both TA and NTA tasks within a single non-auto-regressive (NAR) model. From the modeling perspective, we propose a dual-fusion mechanism to temporally align audio latents with input features for TA tasks, while utilizing cross-attention to integrate input features for NTA tasks, ensuring high-quality generation across both categories. To avoid interference between the two fusion strategies, task-irrelevant features (*i.e.*, NTA features for TA tasks and TA features for NTA tasks) are replaced with learnable dummy embeddings, keeping TA and NTA feature integration disentangled. Both TA and NTA tasks are integrated in each block of the backbone (block-wise fusion), enabling the input to more effectively guide the generation. To balance the amount of different data types, we adopt a task-balanced sampling strategy to balance the ratio between TA and NTA data during training. Moreover, UniFlow-Audio supports a broader range of input modalities than prior works, including text, audio, and visual signals. With all these modalities and tasks involved, UniFlow-Audio learns the shared knowledge across different tasks, which in turn yields competitive or superior performance compared to task-specific baselines. Notably, compared with other unified audio generation models (see Section 2 for details on data and model sizes), our small variant ($\sim$200M parameters), trained on fewer than 8K hours of public data, achieves strong results, underscoring the data efficiency and parameter effectiveness achieved by UniFlow-Audio.

The contributions of this work can be summarized as follows:

1. We provide a novel perspective that formulates diverse audio generation tasks through temporal alignment.

2. We propose UniFlow-Audio, the first flow-matching-based universal audio generation framework that unifies TA and NTA tasks.

3. We design model architectures and data sampling strategies to balance TA and NTA tasks while ensuring the generation quality, including a dual-fusion mechanism, block-wise fusion, and task-balanced sampling.

4. UniFlow-Audio achieves strong results with limited open-source data and parameters on a variety of tasks, demonstrating the advantages of a unified audio generation model.

5. We open-source the code and model to provide a potential unified NAR audio generation foundation model, enabling further theoretical exploration and practical applications.

## 2 RELATED WORK

**Unified Audio Generation** Recently, the research paradigm in audio generation has shifted from task-specific models to unified frameworks capable of handling multiple tasks within a single model. Such frameworks facilitate cross-domain knowledge sharing and improve data efficiency. Representative works include UniAudio (Yang et al., 2024) and AudioX (Tian et al., 2025). UniAudio is a large language model (LLM) based AR model that discretizes audio and various input modalities into token sequences and leverages a multi-scale Transformer to model inter- and intra-frame correlations. UniAudio is trained on 165K hours of data. Despite 11 tasks being included, the video input modality is not supported in UniAudio. In contrast, AudioX adopts an NAR Diffusion Transformer (DiT) with a multi-modal input masking strategy to enhance robustness and generation performance. While trained on 29K hours of large-scale curated data, it focuses exclusively on NTA tasks. Compared with these pioneering works, UniFlow-Audio proposed a flow-matching-based unified NAR framework that achieves good performance on both TA and NTA tasks, with omni input modalities involved (text, audio, video) whilst trained on smaller datasets.

**Flow Matching for Audio Generation** Recent NAR generative models, diffusion models (Ho et al., 2020) and flow matching (Lipman et al., 2023), have attracted significant attention in audio generation due to their strong generative capabilities and the fast inference speed through parallel generation. NaturalSpeech2 (Shen et al., 2024), E3-TTS (Gao et al., 2023), and AudioLDM (Liu et al., 2023) demonstrate the capabilities of latent diffusion models on speech and audio generation. To achieve high-fidelity generation with extremely few steps, flow matching is adopted for T2A and TTS with low latency (Eskimez et al., 2024; Chen et al., 2025; Guan et al., 2024). It alleviates the high inference latency inherent to the iterative denoising process in diffusion models by directly learning a continuous velocity field that transports noise into data in a few integration steps, rather than requiring a substantial number of discrete denoising iterations. Flow matching is also employed in hybrid TTS systems such as CosyVoice (Du et al., 2024) to refine acoustic details given discrete tokens predicted by the AR component. Motivated by the success of flow matching in prior speech and audio generation works, UniFlow-Audio adopts flow matching as the backbone.

## 3 UNIFLOW-AUDIO

As Figure 2 shows, UniFlow-Audio is a unified flow-matching-based audio generation framework that consists of four parts: a variational autoencoder (VAE) that compresses the raw long audio signal into a short sequence, a content encoding part for extracting features from the input content and task instruction, a duration adapter that generates TA content embeddings, and a Transformer-based flow matching backbone.

### 3.1 AUDIO REPRESENTATION FOR GENERATION

Following (Evans et al., 2025), we employ a VAE that operates on raw waveforms for direct waveform generation and reducing latency. The VAE encoder compresses the waveform $\mathbf{x} \in \mathbb{R}^L$ into a latent representation $\mathbf{A} \in \mathbb{R}^{L/2^R \times D}$, where $L$, $R$ and $D$ denote the waveform length, compression ratio and latent dimension, respectively. The VAE architecture also follows (Evans et al., 2025), with details shown in Section G.1. We train the VAE on a mixture of high-quality speech, music, singing voice and general audio datasets to improve the generation performance on various domains.

### 3.2 CONTENT ENCODING WITH TASK INSTRUCTION

All inputs are transformed into continuous embeddings $\mathbf{C}$ instead of discrete tokens to avoid information loss by modality-specific content encoders:

**Phoneme & MIDI**: For TTS, phonemes from grapheme-to-phoneme conversion (g2p)[1] and x-vectors (Wang et al., 2023b) for speaker information are used as input. We use the Transformer-based encoder from FastSpeech2 (Ren et al., 2020) as the content encoder. Singing voice synthesis (SVS) is similar to TTS, except that the input is MIDI rather than phonemes. In addition to phoneme

---

[1]https://github.com/MontrealCorpusTools/Montreal-Forced-Aligner

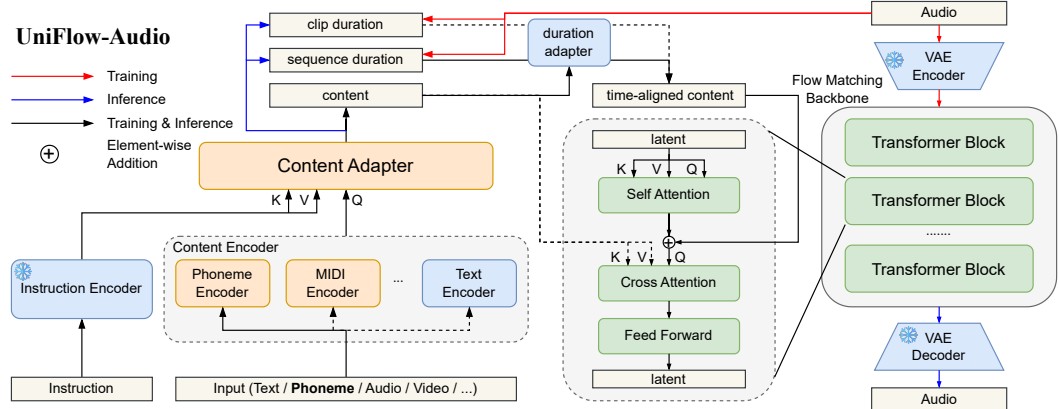

Figure 2: Overview of UniFlow-Audio. The content encoder and adapter transform the input and task instruction into content embedding. Based on the predicted duration, the content embedding is expanded to time-aligned content embedding. A dual-fusion mechanism is applied: the latent is fused with the content by cross attention, and fused with time-aligned content by addition.

embeddings, the MIDI encoder incorporates pitch, pitch duration, and slur information, which are fused with the phoneme embeddings through addition.

**Text**: For T2A and text-to-music generation (T2M), the input is a coarse text description without the alignment information. We use Flan-T5 (Chung et al., 2024) as the encoder following (Majumder et al., 2024; Evans et al., 2025).

**Audio**: For audio input, we reuse the VAE as the encoder to compress the sequence length.

**Video**: For video input in video-to-audio generation (V2A), we use CLIP (Radford et al., 2021) combined as the encoder.

The VAE, Flan-T5, and CLIP are frozen during training. After obtaining $\mathbf{C}$ from the content encoder, we further integrate task instructions to inject explicit task-specific information, enabling the model to distinguish between tasks that share the same input modality (*e.g.*, T2A and T2M). This integration is achieved through an instruction encoder and a content adapter: the former maps the textual instruction into embeddings $\mathbf{I}$, and the latter fuses $\mathbf{C}$ with $\mathbf{I}$ via cross-attention (Attn) and residual connection by

$$\mathbf{C^I} = \text{Attn}(\mathbf{C}, \mathbf{I}, \mathbf{I}) + \mathbf{C}. \tag{1}$$

Regarding each task, we design 10 diverse textual instructions that describe the objective (details shown in Section F). During training, one instruction is randomly selected from each task as the input, whereas during inference, a fixed instruction is used.

With task-involved content embeddings $\mathbf{C^I}$, a clip duration $d_c \in \mathbb{R}^+$ and a sequence duration $d_s \in (\mathbb{R}^+)^L$ are predicted. Since UniFlow-Audio is an NAR model, both TA and NTA tasks rely on $d_c$ to determine the output length. $d_s$ is only required by TA tasks for duration adaptation, which will be introduced in Section 3.3. For the duration predictor, we adopt the architecture in FastSpeech2.

## 3.3 DURATION ADAPTER

As introduced in Section 1, audio generation tasks can be divided into TA and NTA categories by their temporal alignment constraint. In NTA tasks where input and target audio lack temporal correspondence, cross-attention mechanism is typically used to integrate $\mathbf{C}$ into the generation process. In TA tasks, alignment information is often explicitly leveraged for generation. For instance, TTS relies on phoneme-to-frame alignment to expand linguistic units, while speech enhancement (SE) inherently operates on frame-aligned noisy and clean audio pairs. In such cases, content embeddings are aligned and concatenated with audio features, a process that may require a duration adapter.

Building on this insight, we introduce a unified *duration adapter* to explicitly align content embeddings with audio latents across all TA tasks. We posit that this explicit alignment offers superior

efficacy for TA tasks than the implicit mechanisms of cross-attention. Specifically, $\mathbf{C^I}$ is expanded to a time-aligned content $\mathbf{C_T^I}$. That is,

$$\mathbf{C_T^I} = [ \underbrace{c_1^I, \ldots, c_1^I}_{(d_s)_1}, \underbrace{c_2^I, \ldots, c_2^I}_{(d_s)_2}, \ldots, \underbrace{c_N^I, \ldots, c_N^I}_{(d_s)_N} ]. \tag{2}$$

Based on the sequence duration $d_s$, the duration adapter repeats each embedding $c_i^I$ in $\mathbf{C^I}$ for $(d_s)_i$ steps, producing $\mathbf{C_T^I}$ that matches the length of the audio latents. For TTS and SVS, $d_s$ specifies the number of audio latents per phoneme. For SE and V2A, each value in $d_s$ is fixed, since each input audio latent or video frame corresponds to a fixed number of target audio latents. For NTA tasks, $d_s$ is set to a constant dummy value to achieve a unified design. During training, ground-truth durations are used to obtain $\mathbf{C_T^I}$.

## 3.4 DUAL-FUSION FLOW MATCHING TRANSFORMER

The generation backbone is a flow-matching Transformer composed of multiple blocks. Following standard DiTs, we fuse $\mathbf{C^I}$ with the audio latent $\mathbf{A}$ by cross attention in each block. Besides the cross attention, to integrate $\mathbf{C_T^I}$ within each block, we employ a dual-fusion mechanism, where $\mathbf{C_T^I}$ is fused with $\mathbf{A}$ by element-wise addition, as they are temporally aligned. The flow matching timestep $\tau$ is incorporated by adaptive layer norm (AdaLN). Formally, each block contains four operations: a self-attention layer, a cross-attention layer and a feedforward network (FFN) like standard Transformer decoder blocks, with an extra addition fusion between self-attention and cross-attention:

$$\mathbf{A} = (\text{AdaLN}_{\text{SA}} \circ \text{Attn})(\mathbf{A}, \mathbf{A}, \mathbf{A}), \tag{3}$$

$$\mathbf{A} = \mathbf{A} + \mathbf{C_T^I}, \tag{4}$$

$$\mathbf{A} = \text{Attn}(\mathbf{A}, \mathbf{C^I}, \mathbf{C^I}) + \mathbf{A} \tag{5}$$

$$\mathbf{A} = (\text{AdaLN}_{\text{FFN}} \circ \text{FFN})(\mathbf{A}). \tag{6}$$

To prevent interference between the two fusion streams, we replace the ineffective input with learnable dummy embeddings. These embeddings are shared across tasks and initialized as zero vectors.

## 3.5 TRAINING AND INFERENCE

We train the model using the flow matching loss, which encourages the velocity field $v_\theta(\mathbf{z}_\tau, \tau)$ to match a target velocity field, so that the continuous-time flow induced by $v_\theta$ transports data latents $\mathbf{z}_0 \sim p_{\text{data}}$ to a standard Gaussian $\mathbf{z}_1 \sim \mathcal{N}(0, \mathbf{I})$:

$$\frac{d\mathbf{z}_\tau}{d\tau} = v_\theta(\mathbf{z}_\tau, \tau), \quad \mathbf{z}_\tau = (1 - \tau) \cdot \mathbf{z}_0 + \tau \cdot \mathbf{z}_1, \quad \tau \in [0, 1] \tag{7}$$

$$\mathcal{L}_{\text{FM}} = \mathbb{E}_{\tau, \mathbf{z}_0, \mathbf{z}_1} \left\| v_\theta(\mathbf{z}_\tau, \tau, \mathbf{C^I}, \mathbf{C_T^I}) - (\mathbf{z}_1 - \mathbf{z}_0) \right\|^2, \tag{8}$$

where $\theta$ denotes model parameters, $\tau$ is the flow step, and $\mathcal{L}_{\text{FM}}$ is the flow-matching training loss. The two duration predictors are trained together with the backbone using the following losses:

$$\mathcal{L}_{\text{dur-clip}} = \mathbb{E} \| d_c - \hat{d}_g \|^2, \quad \mathcal{L}_{\text{dur-seq}} = \mathbb{E}_i \| (d_s)_i - (\hat{d}_s)_i \|^2, \tag{9}$$

where $\hat{d}_g$ and $\hat{d}_s$ are ground-truth clip duration and sequence duration. For NTA tasks, $\mathcal{L}_{\text{dur-seq}}$ is omitted. In practice, $d_s$ and $\hat{d}_s$ are converted to frame numbers in the logarithmic domain to calculate $\mathcal{L}_{\text{dur-seq}}$, following FastSpeech2. The final training loss is $\mathcal{L} = \mathcal{L}_{\text{FM}} + \mathcal{L}_{\text{dur-clip}} + \mathcal{L}_{\text{dur-seq}}$. During inference, classifier-free guidance (CFG) is employed to balance the trade-off between generated sample diversity and their fidelity to the input content: $v_\theta^{\text{CFG}}(\mathbf{z}_\tau, \mathbf{C}^I, \mathbf{C}_T^I) = v_\theta(\mathbf{z}_\tau, \varnothing, \varnothing) + w \cdot \left( v_\theta(\mathbf{z}_\tau, \mathbf{C}^I, \mathbf{C}_T^I) - v_\theta(\mathbf{z}_\tau, \varnothing, \varnothing) \right)$, where $w$ is the guidance scale.

## 4 EXPERIMENTAL SETUP

**Tasks and Data**    UniFlow-Audio is trained and evaluated on a series of public datasets. Seven tasks are involved: TTS, SVS, T2A, T2M, SE, audio Super Resolution (SR) and V2A. Among them, T2A and T2M are NTA tasks, while the rest are TA tasks. Details of all training and evaluation data are demonstrated in Table 4. A total of 7.7K hours of data are used for training, which is substantially less than that employed in UniAudio and AudioX.

**Task-Balanced Sampling**    As Table 4 shows, different tasks' dataset sizes vary substantially due to discrepancies in collection difficulty and availability. To prevent overexposure to small-scale datasets caused by random sampling, a straightforward approach is to adopt a task-based round-robin sampling strategy: sample data from each task in turn. However, since the number of different task types is imbalanced (five TA tasks and two NTA tasks), task-based round-robin sampling disproportionately favors TA tasks during training, which may in turn affect the model's overall performance. To this end, we upsample data from NTA tasks: T2M by 3 times and T2A by 2 times. We refer to this sampling strategy as *task-balanced sampling*.

**Training**    UniFlow-Audio is trained on eight A100 GPUs with a batch size on each GPU of 24. We train three versions with different sizes: small, medium, and large. Configuration and training details are in Section G.2 and Section G.3. The small version takes about 7 days to train, while the large version takes about 12 days.

**Evaluation Metrics**    For all tasks, both objective and subjective evaluation are conducted. Since UniFlow-Audio is evaluated on a variety of tasks and datasets, we adopt task-specific commonly-adopted metrics, as illustrated in Section B.

## 5 RESULTS

In this section, we first compare the performance of UniFlow-Audio with baselines on all tasks to evaluate the overall generation quality. Then, we explore the effect of CFG scale on different tasks. Finally, we conduct ablation studies on our training and architecture design.

### 5.1 UNIFIED AUDIO GENERATION

The comparison between UniFlow-Audio and prior works is demonstrated in Table 1. For each task, we select a task-specific model from prior works whose architecture and training data are closely aligned with our setting, while also demonstrating competitive performance. Except for LM-based MusicGen (Copet et al., 2023), all other baseline models adopt the diffusion or flow-matching architecture. For F5-TTS, we re-train the model on LibriTTS for 200K steps ($\approx$ 82M training samples) for a fair comparison, as UniFlow-Audio is exposed to 76.8M training samples. UniFlow-Audio achieves at least comparable performance to baselines and significantly outperforms baselines on TTS, SE and SR. For TTS, UniFlow-Audio achieves lower WER with a speaker similarity comparable to F5-TTS. For SVS, a specified vocoder with high reconstruction quality is used in DiffSinger, while UniFlow-Audio uses a universal VAE, resulting in slightly lower singing synthesis quality on soprano samples. For other tasks, UniFlow-Audio performs quite competitively with training only on limited public datasets. In comparison, MusicGen (Copet et al., 2023) was trained on large-scale private datasets.

Previous unified audio generation models, UniAudio (Yang et al., 2024) and AudioX (Tian et al., 2025), are also compared. Despite the difference in the training data, the NAR UniFlow-Audio shows superior generation performance compared with the autoregressive UniAudio. By analyzing speech samples generated by UniAudio, we observe omissions and neglects of words in long sentences, resulting in high WER. This highlights a limitation of AR models: alignment soly based on self-attention can be not robust enough. Here we do not apply post-selection by selecting the sample with the lowest WER from multiple outputs of UniAudio, which was done in the original paper, so such errors happen more frequently. Although post-selection is a common approach for codec-based TTS systems, the inherent instability cased by autoregressive sampling cannot be eliminated.

Table 1: Performance evaluation of UniFlow-Audio and baselines across all tasks.

| Task | Model | Objective Evaluation Metrics | Results | Subjective Evaluation Metrics | Results |
|------|-------|------|------|------|------|
| TTS | F5-TTS (Chen et al., 2025) UniAudio (Yang et al., 2024) UniFlow-Audio | WER↓ \| SIM↑ | 2.93 \| **58.0** 11.93 \| 36.9 **2.19** \| 57.1 | MOS ↑ \| SMOS↑ | **3.79** \| 3.21 |
| SVS | DiffSinger (Liu et al., 2022c) UniFlow-Audio | F0↓ \| SA↑ | **0.144** \| 58.0 0.147 \| **59.9** | MOS↑ \| SMOS↑ | **4.26** \| **4.43** 4.05 \| 4.31 |
| T2A | AudioLDM 2 (Liu et al., 2024b) AudioX (Tian et al., 2025) UniFlow-Audio | FD↓ \| CLAP↑ | 21.8 \| **0.476** 24.7 \| 0.44 **17.2** \| **0.476** | OVL↑ \| REL↑ | **3.57** \| 3.48 3.28 \| 3.33 3.41 \| **3.54** |
| T2M | MusicGen (Copet et al., 2023) AudioX (Tian et al., 2025) UniFlow-Audio | FD↓ \| CLAP↑ | 29.5 \| 0.245 **18.5** \| **0.386** 27.1 \| 0.241 | OVL↑ \| REL↑ | 3.45 \| 3.08 **4.03** \| **3.82** 3.37 \| 3.09 |
| SE | DOSE (Tai et al., 2023) UniAudio (Yang et al., 2024) UniFlow-Audio | PESQ↑ \| STOI↑ | 2.50 \| 0.931 1.77 \| 0.767 **2.91** \| **0.944** | MOS↑ | 3.43 4.02 **4.76** |
| SR | AudioSR (Liu et al., 2024a) UniFlow-Audio | LSD↓ | 1.75 **1.49** | MOS↑ | 3.58 **4.19** |
| V2A | DiffFoley (Luo et al., 2023) AudioX (Tian et al., 2025) UniFlow-Audio | IB↑ \| SYNC↓ | 22.7 \| **922** 28.5 \| 1241 **28.6** \| 1145 | OVL↑ \| SYNC↑ | 2.80 \| 2.94 3.25 \| 3.31 **3.61** \| **3.55** |

AudioX, on the other hand, is restricted to NTA tasks. Thanks to its architectural design, UniFlow-Audio is capable of handling both TA and NTA tasks. By modeling V2A as an NTA task, AudioX can also perform V2A generation. Despite having far fewer trainable parameters, UniFlow-Audio matches or outperforms AudioX on T2A and V2A. AudioX achieves better performance on T2M due to its larger-scale training data.

Table 2: Generation performance across different model sizes.

| Model | # Trainable Params | TTS WER↓ | SVS SA↑ | T2A FD↓ | T2M FD↓ | SE PESQ↑ | SR LSD↓ | V2A IB↑ |
|-------|------|------|------|------|------|------|------|------|
| Prior Works | - | 2.93 | 58.0 | 21.8 | 29.5 | 2.50 | 1.75 | 22.7 |
| UniFlow-Audio small | 208M | 2.27 | 56.6 | 19.7 | **26.2** | 2.60 | 1.58 | 25.5 |
| UniFlow-Audio medium | 395M | **2.10** | 58.4 | 17.8 | 26.6 | 2.72 | 1.53 | 26.5 |
| UniFlow-Audio large | 847M | 2.19 | **59.9** | **17.2** | 27.1 | **2.91** | **1.49** | **28.6** |

We also explore the effect of model size on the generation performance. Table 2 shows that UniFlow-Audio achieves competitive performance even with relatively few parameters. UniFlow-Audio small, with only 208M trainable parameters, already outperforms baseline models across most tasks. This demonstrates that UniFlow-Audio is parameter-efficient, delivering strong results without relying on excessively large model sizes. We assume it can be attributed to the benefit of multi-task training since there is intrinsic commonality in the knowledge required by different tasks. For example, TTS and SVS both require generating vocal from phoneme inputs, while T2A and T2M both require generating sound from coarse textual descriptions. In contrast, other universal generation models, *i.e.*, UniAudio (Yang et al., 2024) and AudioX (Tian et al., 2025), both contain more than 1B parameters. Although medium and large model versions further improve performance on certain tasks, the performance gap between the small model and its larger counterparts remains moderate.

## 5.2 EFFECT OF CFG AND INFERENCE STEPS

We further investigate the impact of two key hyper-parameters in flow matching on generation performance: the guidance scale and the number of inference steps. Interestingly, we observe two

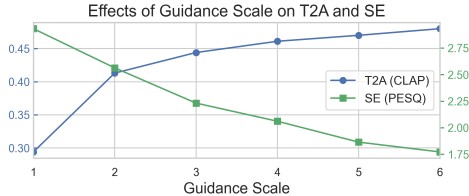 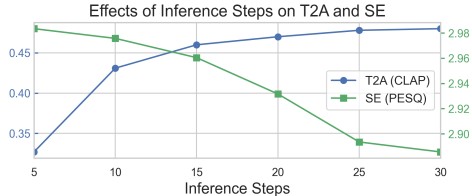

Figure 3: The effect of guidance scale (left) and inference steps (right) on generation performance of typical tasks. When analyzing one factor, the other is kept fixed.

distinct patterns across all tasks: SE and SR fall into one pattern, while the remaining tasks follow another. We take SE and T2A as representative tasks of the two patterns and report their CLAP and PESQ scores, with higher values indicating better performance for both metrics. Results are presented in Figure 3.

For the T2A task, the effects of the guidance scale and inference steps are consistent with typical findings in diffusion-based models: larger guidance scales and more inference steps yield steady performance improvements. This is expected, as stronger guidance provides more effective conditioning from the textual description, while more inference steps allow smaller step sizes in the denoising trajectory, which improves fidelity by reducing error accumulation. However, SE exhibits a sharp performance decline as the guidance scale increases, with PESQ dropping from 2.9 to 1.75. We attribute this to the characteristics of SE: the input inherently contains both signal and noise. Stronger guidance thus amplifies not only the signal but also the noise, leading to reduced perceptual quality in the generated speech. In contrast, the input of T2A is a textual description without "noise", so all information should ideally be reflected in the generated audio. Regarding inference steps, increasing the number of steps is also detrimental to the performance, although the effect is considerably smaller than that of the guidance scale ($2.98 \rightarrow 2.89$). This degradation may also stem from the fact that SE inputs contain both signal and noise. With more inference steps, residual noise can accumulate through the iterative denoising process, slightly reducing the perceptual quality.

## 5.3 Ablation Studies

In this section, we conduct ablation studies to validate several components of UniFlow-Audio: 1) architecture design, including dual-fusion and layerwise fusion mechanisms, and 2) the task-balanced data sampling strategy.

Table 3: Ablation results on the architecture design and data sampling strategies of UniAudio-Flow. The best results are highlighted in bold, while the second-best are underlined.

| Setting | Time Aligned | | | | | Non Time Aligned | |
|---|---|---|---|---|---|---|---|
| | TTS WER↓ | SVS SA↑ | SE PESQ↑ | SR LSD↓ | V2A IB↑ | T2A FD↓ | T2M FD↓ |
| UniFlow-Audio-small | **2.27** | 56.6 | 2.60 | 1.58 | 25.5 | **19.7** | **26.2** |
| w. cross attention | 16.0 | 55.0 | 1.10 | 2.42 | 24.5 | 30.1 | 37.2 |
| w. double fusion | 2.33 | **56.9** | **2.65** | 1.58 | 25.5 | 22.3 | 30.5 |
| w. input fusion | 44.3 | 41.8 | 1.07 | 1.59 | 13.7 | 20.9 | 28.7 |
| w. filler token | 28.3 | 54.4 | 2.50 | 1.62 | 16.6 | 19.8 | 28.5 |
| w/o. balanced sampling | 2.43 | 56.5 | 2.54 | **1.53** | **26.0** | 22.9 | 27.9 |

### 5.3.1 Benefits of Dual-Fusion Transformer

To validate the effectiveness of our proposed dual-fusion mechanism, we replace it with alternative fusion strategies and compare their generation performance. As Figure 4 illustrates, we investigate two alternative fusion mechanisms: *cross-attention fusion* and *double fusion*. Cross-attention fusion

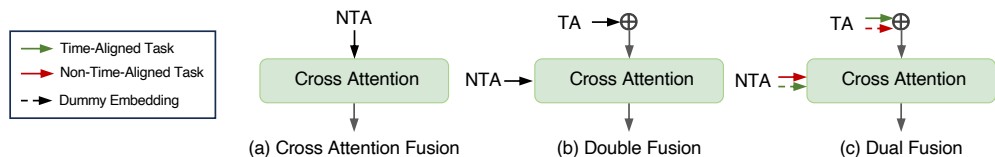

Figure 4: Illustration of different fusion mechanisms, best viewed in color. In the dual fusion sub-figure, green and red indicate the flow in TA and NTA tasks, respectively, while the dashed line represents dummy embeddings. For instance, in TA tasks, the NTA content embeddings are replaced with dummy ones.

is the most straightforward approach, where all contents are fused with the audio latent via cross-attention, similar to AudioLDM2 (Liu et al., 2024b). Double fusion resembles our proposed dual fusion mechanism but differs in one aspect: content embeddings both before and after duration adaptation are fed into the backbone, regardless of the task type. In contrast, in dual fusion, ineffective content embeddings based on task types are set to dummy embeddings. This design may introduce interference between the learning of different task types. In contrast, the dual fusion mechanism employs dummy embeddings, which provide better guidance for the model to attend to different sources depending on the task type, thereby mitigating such interference.

The upper half of Table 3 reports the results of alternative content fusion mechanisms, which are consistent with our assumptions. Although cross-attention has shown strong performance in prior T2A and T2M studies (Liu et al., 2024b), applying it directly to a mixture of task types results in poor performance. Even on non-time-aligned T2A and T2M tasks, its performance is significantly worse than that of dual fusion, suggesting that the presence of rich time-aligned data adversely affects models based on cross-attention. Compared with double fusion, dual fusion achieves similar performance on time-aligned tasks, while substantially outperforming it on non-time-aligned tasks. This demonstrates the effectiveness of the dummy embedding design. As described in Section 3.4, for non-time-aligned tasks, the duration used for content expansion is a dummy value. Consequently, the incorporation of expanded content embeddings into the generation process acts as noise.

### 5.3.2 BENEFITS OF EXPLICIT ALIGNMENT

We investigate the benefits of explicit time alignment by replacing the time-aligned embeddings $C_T^I$ with the original content embeddings $C^I$ padded by filler tokens, following the practice in E2TTS (Eskimez et al., 2024). As shown in Table 3, implicit alignment achieves comparable performance on NTA tasks while TA tasks with one-to-many alignment (see Figure 1) suffer significant performance degradation. This indicates that the one-to-one frame correspondence dominates the alignment learning and leaves less capacity for other TA tasks without explicit alignment. Therefore, explicit alignment is crucial for preserving high-quality generation in tasks with a variety of alignment requirements.

### 5.3.3 BENEFITS OF BLOCK-WISE FUSION

To further validate the architectural design, we examine the effect of fusing time-aligned content embeddings only at the input layer, referred to as *input fusion*. This follows the design of F5-TTS (Chen et al., 2025) and FlowSep (Yuan et al., 2025). As shown in the middle row of Table 3, input fusion leads to a substantial performance drop on time-aligned tasks. Since content embeddings are integrated via cross-attention in each DiT block, injecting time-aligned inputs solely at the input layer makes their influence much weaker than that of non-time-aligned inputs. Consequently, non-time-aligned tasks are only marginally affected, while the performance on time-aligned tasks degrades significantly. In contrast, UniFlow-Audio employs *block-wise fusion*, where time-aligned content embeddings are injected into each DiT block. This progressive fusion allows richer interactions between time-aligned content and audio latents, and proves essential for achieving robust performance across different task types.

### 5.3.4 BENEFITS OF TASK-BALANCED SAMPLING

Finally, we investigate the impact of the proposed task-balanced data sampling strategy. As shown in the last row of Table 3, removing balanced sampling (*w/o balanced sampling*) results in degraded performance on non-time-aligned tasks (T2A and T2M), while performance on time-aligned tasks remain relatively stable. This aligns with the number of datasets from different task types: under the original round-robin sampling strategy, time-aligned tasks are overrepresented. Without explicit balancing, the model is more exposed to time-aligned tasks, which amplifies the influence of time-aligned content input. In contrast, the task-balanced sampling strategy ensures that each task type is adequately represented, mitigating the effects of task imbalance and leading to more consistent and reliable performance across both time-aligned and non-time-aligned tasks.

## 6 LIMITATIONS

Despite unifying TA and NTA audio generation within a flow-matching-based NAR framework, UniFlow-Audio has several limitations. First, tasks involving multiple TA/NTA inputs, such as voice conversion (source speech + target speaker utterance), are not explored. Second, the model's generalization to unseen tasks or input modalities, similar to the zero-shot generalization capabilities of LLMs, has not been investigated. Third, the data and model size have not been scaled. Except for T2M, most tasks have under 1,000 hours of training data. Finally, UniFlow-Audio currently focuses on single-stream audio generation, while multi-stream or multi-source generation (e.g., TTS with background music) remains largely underexplored.

## 7 CONCLUSION

We present UniFlow-Audio, a flow-matching-based universal audio generation framework that unifies both TA and NTA tasks within a single NAR model. By introducing a dual-fusion mechanism with block-wise integration, UniFlow-Audio effectively combines TA and NTA features without cross-task interference. The model leverages shared knowledge across multiple modalities, including text, audio, and vision, to enhance generation performance through unified audio modeling. Extensive experiments demonstrate that, even with limited training data and moderate model size (as small as 200M trainable parameters), UniFlow-Audio achieves competitive performance across diverse tasks, highlighting its potential as a foundation model for unified NAR audio generation.

## ETHICS AND REPRODUCIBILITY STATEMENT

The authors have read and adhere to the ICLR Code of Ethics. This work does not involve human subjects, identifiable private data, or harmful applications. All datasets used are publicly available and were used in accordance with their original licenses and intended purposes. No external sponsorship or conflict of interest influenced the design or conclusions of this work.

All code and source files are provided in the supplementary material and will be publicly released.

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

# A  DATA DETAILS

Table 4: Training and evaluation data details of UniFlow-Audio.

| Task | Training | Evaluation | Training Duration / h |
|------|----------|------------|----------------------|
| TTS | LibriTTS (Zen et al., 2019) | LibriSpeech-PC (Meister et al., 2023) | 555 |
| SVS | M4Singer (Zhang et al., 2022) | | 30 |
| T2A | AudioCaps (Kim et al., 2019) | | 253 |
| SE | LibriTTS+Wham!
VCTK+Wham!
LJSpeech+Musan
VoiceBank+Demand | VoiceBank+Demand
(Botinhao et al., 2016) | 460
44
24
10 |
| SR | HQ-TTS
MUSDB
MoisesDB
FreeSound | VCTK
MUSDB
ESC | 85
47
26
158 |
| T2M | MSD (McFee et al., 2012) | MusicCaps (Agostinelli et al., 2023) | 5789 |
| V2A | VisualSound (Viertola et al., 2025) | | 236 |
| Total | - | | 7717 |

UniFlow-Audio is trained and evaluated on a series of public datasets. Details of all training and evaluation data are shown in Table 4. For TTS, we train the model on LibriTTS (Zen et al., 2019) while use LibriSpeech-PC (Meister et al., 2023) for inference. The cross sentence evaluation setting follows F5-TTS (Chen et al., 2025). For SVS, we use the official training / validation / test splits of M4Singer. Details of other datasets are described in the following:

**T2A**  The official training subset of AudioCaps is used for T2A training. Each sample contains 5 captions in the test subset. Following TANGO (Ghosal et al., 2023), we randomly select one caption per sample for evaluation, and we use the same selected captions as in their setup.

**T2M**  For T2M, we use songs from MSD (McFee et al., 2012) combined with LP-MusicCaps-MSD (Doh et al., 2023) captions as the training data. The original song in MSD can be as long as 14 minutes. During training, we randomly crop 10 seconds for training. The widely-used benchmark MusicGen (Copet et al., 2023) is used for evaluation.

**SE**  For SE, we utilize the method in URGENT challenge (Zhang et al., 2024) to simulate noisy speech. The clean speech datasets include LibriTTS, VCTK Corpus (Yamagishi et al., 2019) and LJSpeech (Ito & Johnson, 2017), while the noise datasets contain WHAM! (Wichern et al., 2019) and noise subset of Musan (Snyder et al., 2015). Room Impulse Rresponses (RIRs) dataset for simulation is the RIRs dataset in Ko et al. (2017). We choose VoiceBank+Demand (Botinhao et al., 2016) for both train and evaluation, which is widely used as a benchmark in SE.

**SR**  For SR, we mainly follow the setup of AudioSR (Liu et al., 2024a), while prioritizing the available sources for ease of collection. The training datasets include MUSDB (Rafii et al., 2019), MoisesDB (Pereira et al., 2023), HQ-TTS (Liu et al., 2022b) and FreeSound (Mei et al., 2024), while the evaluation uses ESC-50 (Piczak, 2015), VCTK-test (Liu et al., 2022a), and MUSDB. All high-quality recordings are first resampled to 24 kHz. Since our VAE is designed to process 24 kHz audio, we choose a cutoff range of [2,6] KHz for the downsampled audio. Based on the method introduced in NVSR (Liu et al., 2022a), we then apply the low-pass filter within this range to simulate low-high resolution audio pairs.

**V2A**  For V2A, since the widely used VGGSound (Chen et al., 2020) dataset is constructed from in-the-wild videos without ensuring high audio-video correspondence, it includes a considerable amount of modality-mismatched samples where the video and audio are not semantically related. This limitation is detrimental to training stability and the inherent irrelevance is harmful to the performance. Therefore, we adopt the smaller but better audio-visual aligned VisualSound (Viertola et al., 2025) for both training and evaluation, which is curated based on ImageBind scores (Girdhar et al., 2023) to identify videos with poor audio-visual correspondence.

## B  EVALUATION METRICS

**TTS**  Following (Wang et al., 2024), we use Word Error Rate (WER)[2] as an objective metric to evaluate the accuracy of generated speech with respect to the given transcription, and Speaker Similarity (SIM)[3] to assess the consistency of speaker characteristics between the generated and prompt speech. For subjective evaluation, we employ the Mean Opinion Score (MOS) to measure overall speech naturalness and the Similarity MOS (SMOS) to assess perceived speaker similarity.

**SVS**  Following Wu et al. (2024), we use root mean square error of fundamental frequency (F0) and semitone accuracy (SA)[4] for objective evaluation. Same as TTS, MOS and SMOS are used as subjective metrics for accessing singing quality and singer similarity.

**T2A & T2M**  Following previous T2A and T2M studies (Liu et al., 2024b), we adopt Frechet Distance (FD) and CLAP score for audio and music generation evaluation. FD measures the similarity of the distribution between generated and reference audio based on PANNs CNN14 (Kong et al., 2020) features, while CLAP score serves as a reference-free metric that captures the semantic alignment between textual descriptions and generated audio.

---

[2]https://huggingface.co/nvidia/stt_en_conformer_transducer_xlarge
[3]https://drive.google.com/file/d/1-aE1NfzpRCLxA4GUxX9ITI3F9LlbtEGP/view
[4]https://github.com/espnet/espnet/blob/master/egs2/TEMPLATE/svs1/svs.sh#L1171

**SE** Following Tai et al. (2023), we choose Perceptual Evaluation of Speech Quality (PESQ) and Short-Time Objective Intelligibility (STOI) for SE evaluation. PESQ measures perceptual speech quality, and STOI estimates speech intelligibility.

**SR** Following previous studies (Liu et al., 2024a; 2022a), we adopt Log-Spectral Distance (LSD) for objective evaluation. LSD measures the discrepancy between the original high-frequency audio and the generated audio. Note that the baseline model AudioSR generates 48 kHz audio, while ours operates at 24 kHz. For fair comparison, AudioSR outputs are downsampled to 24 kHz before evaluation.

**V2A** Following Viertola et al. (2025), we evaluate V2A performance using ImageBind (Girdhar et al., 2023) (IB) and Synchformer (Iashin et al., 2024) (SYNC). IB measures semantic modality consistency by computing the cosine similarity between audio and video embeddings. SYNC assesses synchronization based on temporal offsets between audio and visual modality estimated by Synchformer.

Table 5: Comparison of Multi-Task and Single-Task Training on TTS and T2A.

| Setting | T2A Metrics | | TTS Metrics | |
|---|---|---|---|---|
| | FD ↓ | CLAP ↑ | WER ↓ | SIM ↑ |
| **T2A only** | 25.2 | 0.434 | - | - |
| **TTS only** | - | - | 2.55 | 40.8 |
| **TTS + T2A** | 21.3 | 0.466 | 2.71 | 43.0 |

## C BENEFITS OF MULTI-TASK TRAINING

It is found that multi-task training can improve performance than task-specific training in UniAudio (Yang et al., 2024). Here we also explore the effect of multi-task training on NAR models, taking the combination of TTS and TTA as an example. We perform T2A-only, TTS-only, and TTS + T2A training respectively. For fair comparison, the effective number of TTS / T2A training samples (batch size × training steps) seen by the model are kept the same across the corresponding training configurations. Each task is exposed to the same amount of task-specific training samples across settings. Results are shown in Table 5. It is shown that joint training brings significant improvement in the generation quality. The shared learning objective of generating high-quality audio across tasks enables joint learning to outperform task-specific learning.

## D ANALYSIS OF BLOCK-WISE FUSION DEPTH

To further understand the effect of fusion depth in block-wise fusion, we conduct an ablation study based on the 12-layer small model. Specifically, we remove block-wise fusion in different layer ranges: the first four layers (w.o. early fusion), the middle four layers (w.o. middle fusion), and the last four layers (w.o. late fusion). For efficiency, all models are trained for 200K steps under identical training configurations. The results are summarized in Table 6.

Across all tasks, removing fusion in the middle layers leads to the largest performance degradation, while removing fusion in early or late layers has a relatively smaller impact and even outperforms the original model on some tasks. These findings suggest that the middle layers play a crucial role in integrating time-aligned content. Early layers may focus on low-level acoustic features, primarily capturing patterns from the input audio latent sequence. Late layers may focus on task-specific refinement. They are likely to specialize in task-dependent and domain-specific decoding behaviors (e.g., prosody refinement in TTS or high-frequency compenent refinement in T2M). In contrast, for middle layers, they are at the transition between low-level acoustic encoding and high-level task-specific decoding. They carry semantically enriched but flexible representations, making the contribution of time-aligned fusion in middle layers the most across different layers.

Table 6: Analysis on Block-Wise Fusion Depth.

| | TTS WER↓ | SVS SA↑ | T2A FD↓ | T2M FD↓ | SE PESQ↑ | SR LSD↓ | V2A IB↑ |
|---|---|---|---|---|---|---|---|
| UniFlow-Audio small | 2.27 | 54.5 | 28.0 | 33.9 | 2.41 | 1.66 | 21.3 |
| w.o. early fusion | 2.40 | 56.7 | 26.5 | 32.3 | 2.39 | 1.66 | 22.0 |
| w.o. middle fusion | 2.57 | 55.8 | 28.8 | 32.3 | 2.43 | 1.67 | 20.0 |
| w.o. late fusion | 2.68 | 55.9 | 25.1 | 33.9 | 2.39 | 1.65 | 21.0 |

Table 7: Inference speed comparison between UniFlow-Audio and UniAudio. Reported values indicate inference time per second of audio.

| Model | TTS | SE |
|---|---|---|
| UniAudio | 12.21 | 3.98 |
| UniFlow-Audio small | 0.66 | 0.64 |
| UniFlow-Audio large | 2.23 | 1.26 |

## E  INFERENCE SPEED COMPARISON

To demonstrate the efficacy of UniFlow-Audio, we compare it against UniAudio, reporting the average inference time per second of audio on the same A10 GPU. The results are presented in Table 7. Because of its NAR generation paradigm and smaller parameter size, UniAudio achieves substantially faster inference. In particular, UniAudio-small delivers a ∼18.5× speedup on the TTS task.

## F  TASK INSTRUCTIONS

For each task, we prompt the LLM to generate 10 instructions ranging from simple to complex. These instructions span from basic definitions of the task to detailed specifications of task requirements. Table 8 presents 3 examples of simple, medium, and complex instructions.

## G  ARCHITECTURE & HYPER-PARAMETERS

### G.1  WAVEFORM-BASED VAE

The VAE adopts a fully-convolutional architecture with residual 1D blocks and Snake activations, following the design from Evans et al. (2025). The encoder maps raw waveforms into a compact latent sequence at a downsampling ratio of 480 with 128 channels, while the decoder mirrors the encoder by progressively upsampling the latent sequence with transposed convolution to reconstruct the waveform. To achieve high-fidelity audio generation across different audio types, we train the VAE using a diverse set of datasets from multiple categories, including speech, singing, music, and general audio, with details provided in Table 9. The model is trained for 1M steps on this extensive collection of approximately 6000 hours data, where each audio clip is randomly cropped to 1.5s segments during training.

To measure the reconstruction quality of VAE, we evaluate the mean squared error (MSE) and signal-to-noise ratio (SNR) on held-out test sets. As shown in Table 10, our VAE achieves consistently lower MSE and higher SNR than the one in EzAudio (Hai et al.), which was only trained on AudioSet (Gemmeke et al., 2017).

Table 8: Examples of detailed task instructions.

| TTS | Produce human-like speech from phoneme inputs and speaker representations. |
| --- | --- |
| | Generate natural speech from speaker embeddings and phoneme sequences while maintaining accurate pronunciation. |
| | Convert phoneme sequences into natural speech using speaker embeddings, with precise articulation of words and adaptation to the textual emotional content. |
| T2A | Generate an audio clip based on the given text description. |
| | Synthesize an audio signal from the given text, ensuring the fidelity of sound event representation and the naturalness of the audio output. |
| | Convert the given text into a natural-sounding audio clip, maintaining high fidelity in sound event reproduction (volume, positioning, timing, repetition) and ensuring realistic scene acoustics and event relationships. |
| SVS | Render a singing performance from musical notation, including phonemes, notes, durations, and slurs. |
| | Produce a singing voice rendering derived from the notated score that maintains parametric fidelity to the given phonemes, notes, durations, and slurs. |
| | Synthesize a singing voice that matches the input musica score's specifications (phonemes, notes, durations, slurs) while adapting phoneme durations for natural flow and preserving textual emotional tone. |
| SE | Enhance noisy speech signals by reducing background noise and reverberation. |
| | Improve degraded speech quality by suppressing noise and reverberation while preserving natural voice characteristics. |
| | Enhance speech signals by dynamically suppressing diverse noise types (environmental/mechanical) and reverberation, preserving tonal qualities and timbre across varying SNR conditions. |
| SR | Enhance audio quality by increasing its sampling rate or resolution. |
| | Convert low-sampling-rate audio to high-resolution output, recovering lost high-frequency components and subtle sonic characteristics. |
| | Upsample low-resolution audio signals to higher sampling rates while preserving original signal details and recovering high-frequency components without introducing audible artifacts. |
| V2A | Generate high-fidelity audio synchronized to video. |
| | Produce high-quality audio that matches the video's scene, with accurate timing, spatial positioning, and realistic sound properties. |
| | Generate high-fidelity audio for the video, ensuring strict temporal alignment, correct spatial direction, loudness, and frequency of sounds, while maintaining realism and coherence with visual content. |
| T2M | Develop a music clip that precisely matches the textual description in all aspects. |
| | Produce a musical piece that faithfully represents the given description, incorporating all specified instruments, intended emotions, genre characteristics, and vocal properties. |
| | Generate a musical output that perfectly matches the provided text, incorporating the exact instruments mentioned, upholding authentic stylistic qualities, and delivering the desired emotional impact. If vocals are required, precisely implement the described gender, age, vocal properties, and singing manner. |

Table 9: Datasets used for training the waveform-based VAE.

| Domain | Datasets |
| --- | --- |
| Speech | AISHELL-3 (Shi et al., 2021), TTS-HQ, LJSpeech, LibriTTS, VCTK |
| Singing | OpenSinger (Huang et al., 2021), M4Singer, OpenCpop (Wang et al., 2022), PopCS (Liu et al., 2022c) |
| Music | MUSDB, MoisesDB, MusicCaps |
| General Audio | AudioSet (Gemmeke et al., 2017) |

### G.2 FLOW MATCHING BACKBONE

The diffusion step $\tau$ is processed by a multi-layer perceptron (MLP) to produce AdaLN scale and shift parameters for each Transformer block, conditioning the self-attention and FFN layers:

$$\gamma_{SA}, \beta_{SA}, \alpha_{SA}, \gamma_{FFN}, \beta_{FFN}, \alpha_{FFN} = \text{MLP}(\tau) \tag{10}$$

Table 10: Reconstruction performance of VAE.

| Domain | Speech | | Music | |
|---|---|---|---|---|
| | MSE ↓ | SNR (dB) ↑ | MSE ↓ | SNR (dB) ↑ |
| EzAudio VAE | $4.43 \times 10^{-5}$ | 17.06 | $1.13 \times 10^{-4}$ | 18.09 |
| Ours | $3.84 \times 10^{-5}$ | 17.63 | $8.42 \times 10^{-5}$ | 19.27 |

We apply $\tanh$ to the scaling parameter $\alpha$ in AdaLN (Peebles & Xie, 2023) to improve the numerical stability during training:

$$\mathbf{A}_{\text{norm}} = \gamma \cdot \text{Norm}(\mathbf{A}) + \beta \tag{11}$$

$$\mathbf{A} = \tanh(1 - \alpha) \odot F(\mathbf{A}_{\text{norm}}) + \mathbf{A}_{\text{norm}} \tag{12}$$

To mitigate the potential negative influence from $\mathcal{L}_{\text{dur-clip}}$ and $\mathcal{L}_{\text{dur-seq}}$, we apply gradient scaling to the duration predictors. Specifically, we scale the gradients from the duration losses by a factor $\lambda$ before backpropagation, thereby reducing their influence on the model.

$$\tilde{x} = \lambda \cdot x + (1 - \lambda) \cdot \text{sg}(x)$$

where $\text{sg}(\cdot)$ represents stop gradient operator and $\lambda$ is set to 0.1.

Table 11 summarizes the architectural configurations of different UniFlow-Audio versions. Notably, the small variant contains only approximately 200M trainable parameters, yet it achieves competitive performance as shown in Table 2.

Table 11: Model configurations.

| Model Size | Depth | Embed Size | Num Heads | # Total / Trainable Params |
|---|---|---|---|---|
| Small | 12 | 512 | 8 | 593M / 208M |
| Medium | 16 | 768 | 12 | 780M / 395M |
| Large | 24 | 1024 | 16 | 1.2B / 847M |

### G.3 TRAINING & INFERENCE SETUP

UniFlow-Audio is trained using AdamW optimizer (Loshchilov & Hutter, 2017) with a constant learning rate of 5e-5 with a warmup step of 10K steps and a total training step of 400K steps. To mitigate the negative impact of excessively long audio content sequence on training efficiency, we take a maximum of 5 second audio segments randomly during training for SE and SR. During inference, we take an inference step of 25 by default. Sway sampling (Chen et al., 2025) is adopted to improve the generation performance. During training, both TA and NTA content embeddings are randomly masked with a ratio of 0.2 to train conditional and unconditional generation simultaneously. During inference, a CFG scale of 5.0 is adopted for tasks except SE and SR while CFG is not applied for these two tasks, due to the influence of CFG on them (see Section 5.2).

## H TRAINING STABILITY

We observe frequent loss spikes during training, as shown in Figure 5a, which naturally leads to a question: is training on mixed TA and NTA tasks stable? These loss spikes are typically accompanied by abnormally large loss values (up to 1,000), suggesting that they may be caused by noisy data. Indeed, LP-MusicCaps and VisualSound contain some inherent noise: captions in LP-MusicCaps are generated by ChatGPT and may include hallucinations, while audio-visual correspondence in VisualSound is not guaranteed. To validate this, we exclude LP-MusicCaps and VisualSound (the rest data are denoted as "clean" data) and train the same model. As Figure 5b shows, no spikes occur. Since there are both TA tasks (TTS, SVS, SE, SR) and NTA tasks (T2A) in this setting, we can conclude that training on mixed task types is stable.

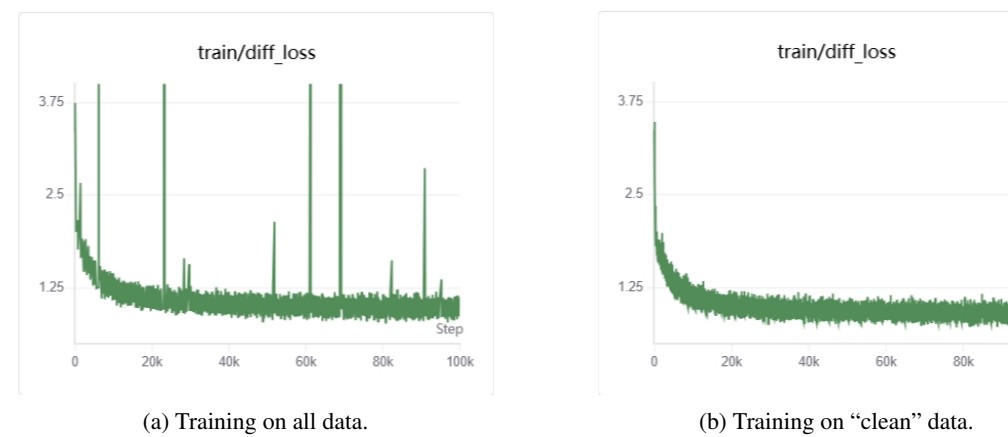

(a) Training on all data.  (b) Training on "clean" data.

Figure 5: Loss curve of training on all data / "clean" data.

## I  LLM USAGE

LLMs were used as assistive tools in this work. Specifically, they were employed to help with limited code writing and debugging, as well as for polishing the language of the paper. The LLMs involved include mainstream models such as GPT, Claude, and Gemini. These model were used for grammar correction, sentence restructuring, and enhancing overall readability. All technical content, experimental design, results, and conclusions were authored and verified solely by the human authors. LLMs did not contribute to the generation of ideas, methods, or data analysis.

## J  SUBJECTIVE EVALUATION DETAILS

For all tasks, we conduct MOS-based subjective tests with explicit instructions for raters. Each sample is rated on a 1–5 Likert scale. We recruit ten raters with college-level education and normal hearing ability for subjective evaluation. Examples of the rating interface and detailed instructions are shown in Figure 6. Below we describe the setup for each task.

For **TTS** and **SVS**, we evaluate speech quality MOS (MOS) and speaker similarity MOS (SMOS). For MOS, raters judge the overall naturalness and listening quality of the synthesized speech or singing voice. For SMOS, raters judge whether the generated audio matches the target/reference speaker in terms of timbre-related characteristics, disregarding prosodic variations.

For **T2A** and **T2M**, we follow AudioGen (Kreuk et al., 2022) and MusicGen (Copet et al., 2023) to evaluate overall quality (OVL) and relevance (REL) to the input caption.

For **SE**, raters assess the intelligibility and naturalness of enhanced speech. Each output is presented together with its clean reference target, and the MOS scores reflect residual noise, processing artifacts, and overall listening quality.

For **SR**, the evaluation setup is identical to SE, except that each sample is additionally accompanied by a spectrogram visualization to facilitate judgments.

For **V2A**, we evaluate overall acceptability (OVL) and synchronization (SYNC) with the reference video. In SYNC evaluation, the raters judge whether audio events are temporally aligned with visual cues such as lip movements, object impacts, or musical actions.

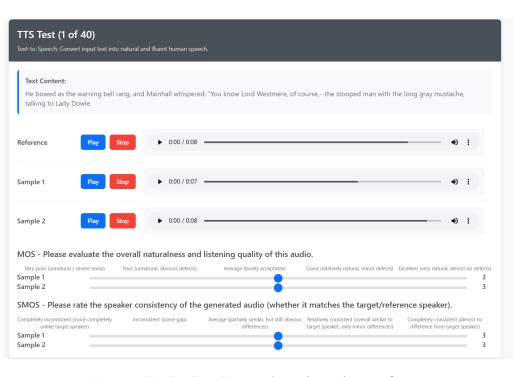

(a) TTS & SVS evaluation interface.

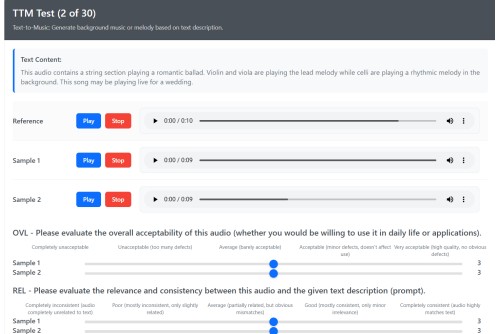

(b) T2M & T2A evaluation interface.

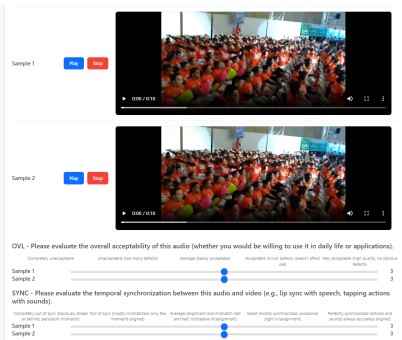

(c) SR evaluation interface.

(d) V2A evaluation interface.

Figure 6: Screenshots of the subjective evaluation interfaces used in our experiments.

