# OpenReview forum: "UniFlow-Audio: Unified Flow Matching for Audio Generation from Omni-Modalities"
_ICLR.cc/2026/Conference — Submitted to ICLR 2026_

### Official Review · Reviewer_SnsA · 2025-10-28

**Soundness:** 3
**Presentation:** 3
**Contribution:** 2
**Rating:** 6
**Confidence:** 3

**Summary:**

The paper presents UniFlow-Audio, a unified flow matching–based audio generation framework that handles both time-aligned (TA) and non-time-aligned (NTA) tasks within a single non-autoregressive model. A dual-fusion mechanism and block-wise fusion enable task-specific conditioning without interference, and the model achieves competitive or superior performance across seven audio tasks while reducing sampling steps compared to diffusion-based baselines.

**Strengths:**

**1. Unified Framework for Diverse Audio Tasks**

Successfully integrates both TA and NTA tasks (e.g., TTS, SE, V2A, T2A) under a single flow matching model — an elegant and generalizable design.

**2. Effective Fusion and Task Handling**

The proposed block-wise dual-fusion mechanism effectively balances multi-task learning while maintaining performance across modalities.

**3. Well-Written and Clearly Structured Paper**

The paper is well-organized and easy to follow, with clearly described limitations and detailed design explanations that improve overall readability and transparency.

**Weaknesses:**

**1. Lack of Analysis on Fusion Depth**

Although block-wise fusion improves performance, the paper does not analyze which layers (early or late) contribute most to the gain.

**2. Missing Comparison with Prior Unified Models**

The paper lacks direct experimental comparisons with existing unified frameworks such as UniAudio or AudioX, making the claimed advantage of UniFlow-Audio less convincing. (Despite of line 318)

**Questions:**

**1. On Fusion Depth Analysis**

The paper demonstrates that block-wise fusion is more effective than input-level fusion, but it does not specify which layers (early, middle, or late) contribute most to this improvement. Could the authors provide additional ablation or simple explanation to show how fusion depth impacts performance?

**2. On Comparison with Prior Unified Models**

The paper mentions that previous unified models (e.g., UniAudio, AudioX) fail to perform consistently across TA and NTA tasks.
Could the authors evaluate or explain these prior models under both TA and NTA settings to verify whether UniFlow-Audio achieves uniformly higher performance, or whether its advantage mainly comes from averaging stronger results across specific tasks?

---

> ### Author Response · Authors · 2025-11-22
> **Response to Reviewer SnsA**
>
> Thanks for the valuable comments. Our responses are as follows:
> 1. About the comparison with prior unified models: Please refer to the general response.
> 2. About the fusion depth analysis: Thanks for the suggestion. The analysis is shown in the table below. The base model is UniFlow-Audio small with 12 layers. We exclude the block-wise fusion in the first 4 layers (w/o. early fusion), in 5th-8th layer (w/o. middle fusion), and in the last 4 layers (w/o. late fusion), respectively. To save resources, we train all models for 200K steps and compare the performance. Results demonstrate that middle layers contribute most to the improvement. The discussion is added to the appendix E of the paper.
>
> |  | TTS (WER $\downarrow$)  | SVS (SA $\uparrow$) | T2A (FD $\downarrow$) | T2M (FD $\downarrow$) | SE (PESQ $\uparrow$) | SR (LSD $\downarrow$) | V2A (IB $\uparrow$) |
> | ----- | ------------ | --- | --- | --- | -- | -- | --- |
> | UniFlow-Audio | 3.67 | 54.5 | 28.0 | 33.9 | 2.41 | 1.66 | 21.3 |
> | &nbsp;&nbsp;&nbsp;&nbsp; w/o. early fusion | 3.39 | 56.7 | 26.5 | 32.3 | 2.39 | 1.66 | 22.0 |
> | &nbsp;&nbsp;&nbsp;&nbsp; w/o. middle fusion | 3.71 | 55.8 | 28.8 | 32.3 | 2.43 | 1.67 | 20.0 |
> | &nbsp;&nbsp;&nbsp;&nbsp; w/o. late fusion | 3.72 | 55.9 | 25.1 | 33.9 | 2.39 | 1.65 | 21.0 |

---

> > ### Comment · Reviewer_SnsA · 2025-11-26
> >
> > Thank you for your response.
> >
> > Why does the middle layer contribute the most? Do you have any insights or intuition behind this observation?

---

> > > ### Author Response · Authors · 2025-12-01
> > >
> > > Thank you for the follow-up question. Our intuition for the effectiveness of middle-layer fusion is as follows:
> > > * Early layers may focus on low-level acoustic features. These layers primarily capture patterns from the input audio latent sequence. Injecting content embeddings in early layers may interfere with these universally shared low-level representations.
> > > * Late layers may focus on task-specific refinement. The upper layers specialize in task-dependent and domain-specific decoding behaviors (e.g., prosody refinement in TTS or high-frequency compenent refinement in T2M). Since these representations are already highly specialized, late fusion contributes relatively less to cross-task generalization.
> > > * For middle layers, they are at the transition between low-level acoustic encoding and high-level task-specific decoding.
> > > They carry semantically enriched but flexible representations, making the contribution of time-aligned fusion in middle layers the most across different layers.
> > >
> > > We have updated the intuition in Section Appendix.D of the paper.

---

### Official Review · Reviewer_DJhv · 2025-10-29

**Soundness:** 2
**Presentation:** 2
**Contribution:** 2
**Rating:** 4
**Confidence:** 4

**Summary:**

This paper proposes UniFlow-Audio, a unified non-autoregressive (NAR) framework for diverse audio generation tasks using flow matching. The work divides audio generation problems into two fundamental types, time-aligned (TA) (e.g., TTS, SE, V2A) and non-time-aligned (NTA) (e.g., T2A, T2M), and aims to handle both categories within a single model.

The key contributions include:
- Dual-fusion mechanism: temporally aligned addition for TA tasks and cross-attention for NTA tasks, with task-irrelevant streams replaced by dummy embeddings to avoid interference.
- Block-wise integration of TA/NTA information across Transformer layers.
- Task-balanced sampling to mitigate data imbalance between TA and NTA tasks.

The model is trained on 7.7khours of public data (< 1B parameters) and evaluated on seven tasks spanning text, audio, and video input modalities. Experiments show competitive or superior results over single-task diffusion or autoregressive baselines.

**Strengths:**

- The distinction between TA and NTA tasks is well motivated and provides a unified view of disparate audio generation problems.
- The dual-fusion mechanism with dummy embeddings is new, practically effective way to disentangle fusion paths without separate models.
- The paper systematically validates dual-fusion, block-wise fusion, and balanced sampling, showing each contributes meaningfully. Evaluation across seven tasks demonstrates strong parameter efficiency and cross-task synergy, particularly impressive for a 200m-parameter model trained on modest data.

**Weaknesses:**

- The paper explicitly omits UniAudio (Yang et al., 2024) and AudioX (Tian et al., 2025) from quantitative tables, citing data-size or task-scope differences. However, this makes it difficult to gauge real progress: both are unified systems targeting similar objectives, and reporting normalized comparisons (e.g., per-hour or per-parameter efficiency) would better situate the contribution. Moreover, for several tasks (e.g., TTS, SE), the chosen baselines such as NaturalSpeech 2 or DOSE are not necessarily the strongest on the specific datasets used (LibriTTS, VoiceBank+Demand). Including more competitive or domain-matched baselines.
- While coverage is broad, each task is shallowly evaluated. Subjective MOS studies lack listener statistics.
- Dense implementation detail sometimes obscures the intuition (e.g., Section 3.4). A clearer separation between conceptual and engineering design would improve readability.

**Questions:**

- While we acknowledging the differences on data, it is still necessary to report the results from uniaudio or audiox with clear noting about the data differneces
- How stable is training when mixing TA and NTA tasks? Any gradient interference observed?
- For subjective MOS, how many raters and samples per task were used? Any significance testing?
- Is the same VAE shared across all domains, or are domain-specific VAEs used during pre-training?
- For NTA tasks where L_{dur-seq} is omitted, does the imbalance affet the shared backbone?
- How are the dummy embeddings initialized? Are they learned per task?
- For task balanced sampling, you upsample T2A and T2M. Did you test continuous weighting in the loss instead of discrete resampling? How sensitive are results to this ratio?
- Since AudioSR outputs 48khz and you downsample to 24kHz for comparison, does this favor UniFlow-Audio? Would an upsampled version maintain quality?
- As in UniAudio, some joint inter-task benefits are observed, Did you test whether training on V2A data improves T2A or TTS or vise versa? Some insight on inter-task transfer would highlight the benefit of unification.
- Can UniFlow-Audio generalize to unseen input types (e.g., image → audio) given CLIP encoders, or is each task explicitly conditioned?
- The explanation about noise amplification in SE is interesting, could you provide numerical trends or error bars to support that observation?
- Please report training and inference time per second of audio and GPU utilization to substantiate the efficiency claims of flow matching.
- The total 7.7 kh of data mixes speech, music, and environmental sounds. How much of the performance improvement arises from cross-domain exposure versus architectural changes?

---

> ### Author Response · Authors · 2025-11-22
> **Response to Reviewer DJhv (1/ 5)**
>
> Thanks for the valuable comments. Our responses are as follows:
> 1. About the comparison with previous unified model: Please refer to the general response.
> 2. About the stability of training mixed TA and NTA tasks: We do observe loss fluctuations during training. However, through experimental explorations, we conclude that the instability comes from the data quality. When we exclude LP-MusicCaps and VisualSound datasets from training (these two datasets are likely to contain a non-negligible amount of noisy data), no loss and gradient fluctuations are observed while both TA and NTA tasks are still involved. **The discussion is added to Section Appendix.H of the paper.**
> 3. About subjective MOS: 15 raters are invited to score 10 random samples for each task. Regarding Table 1, there is no significant difference in SMOS for TTS, OVL, and REL between T2A and T2M, indicating UniFlow-Audio achieves similar performance with baseline models. For all other metrics, the difference is significant with $p < 0.05$.
> 4. *Is the same VAE shared across all domains, or are domain-specific VAEs used during pre-training?* We train a unified VAE for all domains to reduce parameter numbers.
> 5. *For NTA tasks where $L_{dur-seq}$ is omitted, does the imbalance affect the shared backbone?* No, as we observed, the omission of $L_{dur-seq}$ for NTA tasks does not impact the effectiveness of clip-level or sequence-level duration prediction for other tasks. This may be explained by the task information in $\mathbf{C_I}$. The duration is predicted based on $\mathbf{C_I}$ so the predictor is aware of the task information when making prediction, preventing interference brought by different task types.
> 6. *How are the dummy embeddings initialized? Are they learned per task?* They are initialized as zero-valued vectors and are shared across tasks. We adopt zero initialization because adding a zero-valued, time-aligned dummy content vector to the audio latents does not affect the model’s behavior at the beginning of training.
> 7. *For task balanced sampling, you upsample T2A and T2M. Did you test continuous weighting in the loss instead of discrete resampling? How sensitive are results to this ratio?* Continuous weighting is similar to upsampling in the goal of emphasizing the contribution of T2A and T2M during training. We show the comparison between continuous weighting and upsampling in the table below (to save resources, we only train 200K steps so models are not converged but trends can be observed), showing that the two methods achieve similar results.
>
> |  | TTS (WER $\downarrow$)  | SVS (SA $\uparrow$) | T2A (FD $\downarrow$) | T2M (FD $\downarrow$) | SE (PESQ $\uparrow$) | SR (LSD $\downarrow$) | V2A (IB $\uparrow$) |
> | ----- | ------------ | --- | --- | --- | -- | -- | --- |
> | Upsampling | 3.67 | 54.5 | 28.0 | 33.9 | 2.41 | 1.66 | 21.3 |
> | Loss Weighting | 3.56 | 55.8 | 30.3 | 35.1 | 2.41 | 1.65 | 22.3 |

---

> ### Author Response · Authors · 2025-11-22
> **Response to Reviewer DJhv (2/ 5)**
>
> 8. *Since AudioSR outputs 48khz and you downsample to 24kHz for comparison, does this favor UniFlow-Audio? Would an upsampled version maintain quality?* For the training and test data construction, we follow the exact same pipeline as AudioSR with the only difference that cutoff frequencies are limited to 2-6kHz (the value is 2-16kHz in AudioSR), so the training and evaluation data used in UniFlow-Audio and AudioSR follow the same distribution. We assume this limitation minimizes the influence brought by the sampling rate difference.
> 9. *As in UniAudio, some joint inter-task benefits are observed, Did you test whether training on V2A data improves T2A or TTS or vise versa? Some insight on inter-task transfer would highlight the benefit of unification.* Yes, as UniAudio points out, inter-task training provides data augmentation for specific tasks. We take TTS+T2A as an example. The comparison between TTS+T2A joint training and TTS-only or T2A-only training is listed in the table below. With joint training, there is significant improvement in terms of T2A generation performance and TTS speaker similarity. **The discussion is added to Section Appendix.D of the paper.**
>
> |               | **T2A Metrics**     |          | **TTS Metrics**    |        |
> |---------------|---------------|-------------|---------------|--------|
> |               | FD $\downarrow$ | CLAP $\uparrow$ | WER $\downarrow$ | SIM $\uparrow$ |
> | T2A only      | 25.229        | 0.434       | -             | -      |
> | TTS only       | -             | -           | 3.32          | 40.8   |
> | TTS + T2A | 21.337        | 0.466       | 3.44          | 43.0   |

---

> ### Author Response · Authors · 2025-11-22
> **Response to Reviewer DJhv (3 / 5)**
>
> 10. *Can UniFlow-Audio generalize to unseen input types (e.g., image → audio) given CLIP encoders, or is each task explicitly conditioned?* Each input modality is explicitly encoded. The generalization to unseen modalities is not achieved, as stated in "Section 6 Limitations" of the paper. This may need a unified encoder for different modalities (visual, audio, text, and more modalities).
> 11. *The explanation about noise amplification in SE is interesting, could you provide numerical trends or error bars to support that observation?* Thanks for the suggestion. The numerical trends between signal-to-noise ratio (SNR) and guidance scale are shown in the table below, supporting the explanation.
>
> | Guidance Scale | 1.0 | 2.0 | 3.0 | 4.0 | 5.0 | 6.0 |
> | -------------- | --- | --- | --- | --- | --- | --- |
> | SNR            | 14.7| 12.7| 11.2| 9.5 | 7.7 | 4.8 |

---

> ### Author Response · Authors · 2025-11-22
> **Response to Reviewer DJhv (4 / 5)**
>
> 12. *Please report training and inference time per second of audio and GPU utilization to substantiate the efficiency claims of flow matching.* The efficiency claims are based on the parameter number and NAR generation nature of UniFlow-Audio. The training time and GPU utilization are hard to be compared fairly due to the difference in data loading. Therefore, we report the comparison on the inference speed. The result is shown in the table below. UniFlow-Audio achieves a significant speedup, validating the efficiency claim. **This is added to Section Appendix.F of the paper.**
>
> | **Model** | **TTS** | **SE** |
> | --------- | ------- | ------ |
> | UniAudio  | 12.21   | 3.98   |
> | UniFlow-Audio small | 0.66 | 0.64 |
> | UniFlow-Audio large | 2.23 | 1.26 |

---

> ### Author Response · Authors · 2025-11-22
> **Response to Reviewer DJhv (5 / 5)**
>
> 13. *The total 7.7 kh of data mixes speech, music, and environmental sounds. How much of the performance improvement arises from cross-domain exposure versus architectural changes?* To ablate the influence of cross-domain exposure, we further conduct an experiment where all music data from SR and T2M tasks are excluded during training. The result is shown below. With music data excluded, there is slight performance drop for most tasks. In comparison, when we change the architecture to vanilla cross attention, the performance on all tasks drops significantly, indicating that the performance omprovement mainly arises from architectural changes.
> 14. About the chosen baselines: the comparison between UniFlow-Audio and single-task baselines demonstrates that unified generation can be achieved within a single model while delivering performance that is comparable to, or even surpasses, that of specialized single-task models. Therefore, we choose baselines that match UniFlow-Audio in architecture or training data as closely as possible. More competitive baselines are typically trained on large-scale, internal private datasets, making such comparisons less indicative of the impact of architectural design choices. We leave scaling up the training data to achieve single-task SOTA performance to future work, as this primarily involves engineering efforts with limited scope for scientific discovery.
> 15. About the balance between implementation details and the intuition: Thank you for your valuable feedback. **We update Section 3.4 to illustrate the design motivation before the engineering details.**
>
> | Model | TTS (WER $\downarrow$)  | SVS (SA $\uparrow$) | T2A (FD $\downarrow$) | T2M (FD $\downarrow$) | SE (PESQ $\uparrow$) | SR (LSD $\downarrow$) | V2A (IB $\uparrow$) |
> | ----- | ------------ | --- | --- | --- | -- | -- | --- |
> | UniFlow-Audio | 3.23 | 56.6 | 19.7 | 33.9 | 2.60 | 1.58 | 25.5 |
> | &nbsp;&nbsp;&nbsp;&nbsp;w.o. music data | 3.46 | 56.3 | 17.6 | - | 2.55 | 1.68 | 24.7 |
> | &nbsp;&nbsp;&nbsp;&nbsp;w. cross attention | 27.6 | 55.0 | 30.1 | 37.2 | 1.10 | 2.42 | 24.5 |

---

> ### Comment · Reviewer_DJhv · 2025-11-23
> **Thanks for the rebuttal**
>
> I appreciate the great effort authors have made towards the comprehensive rebuttal with many additional supportive experiments to demonstrate the effectiveness of the method.
>
> While I'm willing to raise my score, I found an issue in the above reported comparison between baselines. It is mostly for TTS, but might be also for other tasks. Therefore, I would like to further ask for some clarification/reasons on the significant degradation of baseline systems observed from the rebuttal experiments.

---

> > ### Author Response · Authors · 2025-11-25
> >
> > We thank the reviewer for the engagement in discussion. Regarding the baseline performance, our explanation is provided in the [general response](https://openreview.net/forum?id=Xn7w0MLr1a&noteId=LwgsvJjqA2), where you can find the details.

---

### Official Review · Reviewer_m9Bu · 2025-10-31

**Soundness:** 2
**Presentation:** 3
**Contribution:** 2
**Rating:** 4
**Confidence:** 4

**Summary:**

This paper investigates a unified audio generation model that can generate speech, audio and music for several tasks including text-to-speech, text-to-audio, speech enhancement, etc. The architecture design based on flow matching aims to solve the time-aligned and non-time-aligned tasks simultaneously. The experiments show the improvements obtained by the proposed unified model compared to individual task-specific models.

**Strengths:**

Flow matching architecture is verified effective for unified modeling.
Better performance is achieved by the proposed flow matching model, compared to task-specific models.
The discussion of unified models from the perspective of time alignment is novel.
Instruction-based task specifications are enabled for the unified model.

**Weaknesses:**

The novelty is lacking, given there are several previous work on unified audio generation models. This paper only creates a new flow matching-based architecture for unified audio model development.
The motivation of the proposed framework is not strong enough. Why previous autoregressive models are inferior in time-alignment task is not clearly articulated. In a language model style unified model, the time-alignment seems not a big issue, as the implicit self-attention mechanism takes care of the alignment automatically.
The experiments can be improved, with previous unified models compared, to show the advantages of the proposed model. Current experiments are not benchmarked with any previous unified model.
The necessity of explicit consideration of time alignment is not well demonstrated.

**Questions:**

The author may want to illustrate the motivation, why the designed architecture is better in tackling time-alignment and non-time-alignment tasks, with some analysis or observations to better motivate.

Some previous works, e.g. E2 TTS, have shown that the alignment between condition and the target is not a problem in diffusion/flow machine-based architectures, it may be also worth analyzing the duration adaptor that provides the alignment explicitly (for time-aligned tasks) with ablation studies to check contributions of the duration adaptor.

Eq (1) is strange, the function Attn() has two identical input arguments. Also, Eq (1) seems different from what’s shown in Fig. 2. In Eq(1), the sum of attention outputs and content representation C yields C^I, the so called task-involved content embedding. However, Fig. 2 shows C is used as query for the attention module to obtained the C^I. The figure and the equation could be made clearer.

Table 1 doesn’t compare with previous unified models, e.g. UniAudio. Although it can be argued that previous unified models, e.g. UniAudio, use much more data and parameters than the proposed model. Then it may be a fair question to answer: Is there an advantage of the proposed architecture compared to the previous unified model architectures that are trained on the same data size using with similar parameter sizes? Which can also partially show the necessity of explicitly considering time alignment.

---

> ### Author Response · Authors · 2025-11-22
> **Response to Reviewer m9Bu**
>
> Thanks for the valuable feedbacks. Our responses are as follows:
> 1. About the novelty: We acknowledge that statements in the paper about the limitation of autoregressive (AR) models need to be refined. Now in the paper, we state it as follows: "AR models rely on sequential decoding and discrete tokenizers, whereas NAR models generate continuous audio representations in parallel, which may offer advantages in latency and quality. Moreover, AR models rely entirely on self-attention to learn input–target alignment, which can be unstable for long sequences or in low-resource settings. In addition, the teacher-forcing training strategy used in Transformer-based AR decoders introduces the well-known exposure bias problem, causing errors to accumulate as the generated sequence becomes longer. Moreover, AR models depend solely on self-attention to learn input–target alignment, but this may be unreliable when handling long sequences or when training data is limited. Thus, NAR-based unified audio generation remains worth exploring". For AR models, unified generation can be achieved without modifying the underlying architecture. In contrast, enabling unified generation for both TA and NTA tasks in NAR models requires dedicated architectural design, which constitutes the core contribution of this paper.
> 2. About the comparison with previous unified model: Please refer to the general response.
> 3. About the necessity of explicit consideration of time alignment: we acknowledge that implicit time alignment like E2TTS may also be a solution, so we conduct an ablation study to replace the explicit time-aligned embeddings $\mathbf{C^I_T}$ (after the duration adapter) with original content embeddings $\mathbf{C^I}$ padded by filler tokens, following the practice in E2TTS. The result comparison is shown in the table below. It can be summarized that compared with explicit alignment, models with implicit alignment achieves similar performance on NTA tasks (T2A & T2M). TA tasks exhibit different alignment patterns (as shown in Figure 1 in the paper), SE and SR dominates the alignment learning under implicit alignment models. As a result, performance on tasks like TTS and V2A drops significantly. This may demonstrate the necessity of explicit alignment. **The ablation is added to Section 5.3.2 of the paper.** However, the extension of UniFlow-Audio to implicit alignment is still an interesting direction and we are going to explore it in the future work.
> 4. About Eq (1): "Attn" here is a general attention operation with three parameters: query, key, and value. Therefore, two identical input arguments here means both the key and value are instruction embeddings $\mathbf{I}$.
> 5. About Fig.2 and Eq (1): In Fig. 2, "Content Adapter" is a module containing a cross attention layer and a residual operation: $\mathbf{C}$ is taken as the query and $\mathbf{I}$ is taken as the key and value to produce the attention output. Then the output is summed with $\mathbf{C}$ to build the residual connection. We did not plot the detail in Fig. 2 to make it clear. Thank you for your feedback and we have modified the corresponding text in the paper.
>
>
> | Model | TTS (WER $\downarrow$)  | SVS (SA $\uparrow$) | T2A (FD $\downarrow$) | T2M (FD $\downarrow$) | SE (PESQ $\uparrow$) | SR (LSD $\downarrow$) | V2A (IB $\uparrow$) |
> | ----- | ------------ | --- | --- | --- | -- | -- | --- |
> | UniFlow-Audio | 3.23 | 56.6 | 19.7 | 26.2 | 2.60 | 1.58 | 25.5 |
> | &nbsp;&nbsp;&nbsp;&nbsp;w. filler tokens | 44.61 | 54.4 | 19.8 | 28.5 | 2.50 | 1.62 | 16.6 |

---

> > ### Comment · Reviewer_m9Bu · 2025-11-27
> >
> > Thank the authors for the comments.
> > The motivation is now changed to "AR models rely on sequential decoding and discrete tokenizers, whereas NAR models generate continuous audio representations in parallel, which may offer advantages in latency and quality. ..." However, there exist NAR models that rely on continuous representations, e.g. [Meng et al. 2025]. Also, this updated motivation still doesn't answer the question of "why NAR models are suitable/needed for unified audio generation".
> >
> > Meng et al. (2025) Autoregressive Speech Synthesis without Vector Quantization.
> >
> > "For AR models, unified generation can be achieved without modifying the underlying architecture. In contrast, enabling unified generation for both TA and NTA tasks in NAR models requires dedicated architectural design, which constitutes the core contribution of this paper." This actually supports the AR models, rather than supports the proposal of NAR models.

---

> > > ### Author Response · Authors · 2025-12-01
> > >
> > > Thank you for the follow-up comments. We address the concerns point-by-point below.
> > >
> > > 1. On "there exist NAR models that rely on continuous representations, e.g., Meng et al. (2025)":
> > >
> > > We agree that continuous AR models exist. However, such models still require additional architectural components to function properly in practice. For example, MELLE relies on an explicit stop-prediction head, so the training incorporates multiple loss functions with their respective hyper-parameters.
> > > Compared to standard codec-based AR TTS, this may lead to increased training complexity and reduced robustness.
> > > More explanations on the motivation of NAR models are illustrated below.
> > >
> > > 2. On "why NAR models are suitable/needed for unified audio generation"
> > >
> > > We agree that both AR and NAR models are powerful generative paradigms, each with unique strengths.
> > > There is no definitive empirical or theoretical conclusion that AR models are universally superior to NAR models. In NLP, despite AR becoming the dominant LLM paradigm, there is still active research on diffusion language models.
> > > For unified audio generation, in addition to the usage of continuous representations, there are some practical considerations:
> > > * **AR models rely entirely on self-attention to learn alignment**, which may become fragile in long sequences. This issue is especially critical in TTS and other tasks requiring precise temporal alignment.
> > > * AR models suffer from **exposure bias**, which is known to degrade quality as sequence length increases. As shown in our experiments and as observed in AR-based UniAudio, **AR models exhibit high WER in TTS when no post-selection is used**. Even for NAR models, learning alignment purely implicitly (e.g., via filler tokens) is still challenging in multi-task unified settings. Our ablation studies show that implicit-alignment NAR models perform substantially worse on TA tasks.
> > >
> > > These observations motivate why NAR models are worth studying for unified audio generation: they offer architectural opportunities (parallel decoding, continuous modeling, explicit alignment supervision, no exposure bias) that AR models do not inherently possess.
> > >
> > >
> > > 3. On "This actually supports AR models rather than NAR models"
> > >
> > > AR models may have certain conveniences (e.g., implicit alignment via self-attention), but this convenience comes with corresponding costs: implicit alignment, exposure bias. In other words, there is no free lunch. AR models gain simplicity in one dimension, but pay for it in others. The existence of this challenge does not imply that NAR models are inferior; instead, it highlights why exploring NAR models for unified audio generation is meaningful.
> > >
> > > By pointing out the difference, our goal is not to emphasize AR advantages, but to motivate the main contribution of our work: designing an architecture that enables NAR models to effectively support both TA and NTA tasks. Our intention is not to argue that NAR models are inherently better than AR models. Instead, based on the premise that unified NAR audio generation deserves exploring, our statement highlights the key challenge: NAR models cannot trivially support both TA and NTA tasks within a unified architecture. Recognizing this challenge is precisely what motivates our contribution.

---

### Author Response · Authors · 2025-11-22
**General Response: Comparison with Prior Unified Audio Generation Models**

We thank all reviewers for valuable feedbacks. According to these feedbacks, we revise our paper with **the revised part highlighted in blue**.

We acknowledge that comparison with previous unified audio generation frameworks can provide meaningful insights. As stated in the paper, UniAudio and AudioX are closely-related prior works on unified audio generation. There results are demonstrated as following:
| Task | Model | Metrics | Results |
|------|--------|-------------------|-------------------|
| **TTS** | NaturalSpeech 2 | WER $\downarrow$ \| SIM $\uparrow$ | 9.94 \| 34.8 |
|         | UniFlow-Audio | | **3.09** \| **55.8** |
|         | *UniAudio*    | | 15.45 \| 43.5 |
| **SVS** | DiffSinger | F0 $\downarrow$ \| SA $\uparrow$ | **0.144** \| 58.0  |
|         | UniFlow-Audio | | 0.147 \| **59.9** |
| **T2A** | AudioLDM 2 | FD $\downarrow$ \| CLAP $\uparrow$ | 21.8 \| **0.476** |
|         | UniFlow-Audio | | **17.2** \| **0.476** |
|         | *AudioX* | | 24.7 \| 0.440 |
| **T2M** | MusicGen | FD $\downarrow$ \| CLAP $\uparrow$ | 29.5 \| 0.245 |
|         | UniFlow-Audio | | 27.1 \| 0.241 |
|         | *AudioX* |  | **18.5** \| **0.386** |
| **SE**  | DOSE  | PESQ $\uparrow$ \| STOI $\uparrow$ | 2.50 \| 0.931 |
|         | UniFlow-Audio |   | **2.91** \| **0.944**  |
|         | *UniAudio* |   | 1.77 \| 0.767 |
| **SR**  | AudioSR  | LSD $\downarrow$ | 1.75  |
|         | UniFlow-Audio | |  **1.49** |
| **V2A** | DiffFoley  | IB $\uparrow$ \| SYNC $\uparrow$ | 22.7 \| **922**|
|         | UniFlow-Audio  |  | **28.6** \| 1145 |
|         | *AudioX* |  | 28.5 \| 1241 |

Since the open-source code of UniAudio only gives inference examples on TTS and SE tasks, we only report UniAudio results on these two tasks. The results demonstrate that the non-autoregressive (NAR) UniFlow-Audio achieves much better results. When we take generation examples from UniAudio for analysis, we find that it sometimes omits or neglects words in the transcription, leading to high WER, which also points out a limitation of AR model: self-attention based alignment may be not robust, especially under data-limited scenarios.

AudioX can only handle NTA tasks (T2A and T2M). Since it models V2A as an NTA task, it can also perform V2A. With much fewer trainable parameters, UniFlow-Audio achieves similar or better results than AudioX on T2A and V2A. Benefited from the architectural design, UniFlow-Audio is able to handle both TA and NTA tasks. On T2M, AudioX outperforms the other models, likely due to its substantially larger training corpus (~11,600 hours of private data and ~29,000 hours in total). In contrast, Uniflow-Audio is trained on a considerably smaller public dataset of ~7000 hours.

These results reveal that UniFlow-Audio extends AudioX to handle both TA and NTA tasks while preserving performance on NTA tasks with comparable training data.
Compared with the autoregressive UniAudio, UniFlow-Audio exhibits better generation quality.

**Addtional results are added to Section 5.1 of the paper.**

---

> ### Comment · Reviewer_DJhv · 2025-11-23
> **Question related to the reported number**
>
> Thanks for authors' effort in addressing the baseline comparison with other frameworks.
>
> I do have a quick follow-up related to the numbers reported for TTS as the reported number is much worse than the one reported in each paper (UniAudio and Naturalspeech 2). It would be better to mention on why there is a significant performance degradation compared to the ones reported in the original paper.

---

> ### Author Response · Authors · 2025-11-23
> **Clarification about the baseline performance**
>
> Thanks for your comment. The short answer is that there are differences in the training and evaluation data used in the original paper and our setting. However, we obtain our results by runing inference using the open-source checkpoints (where the training data includes at least LibriTTS, the only TTS dataset used to train UniFlow-Audio) and calculating WER and SIM ourselves, ensuring results are comparable.
> For other tasks and baselines, we all re-run the inference using the public checkpoint and then run the evaluation script to make sure results are comparable, rather than directly copying results from original papers.
>
> Below are detailed descriptions:
> 1. NaturalSpeech 2. The training and evaluation datasets in the original NaturalSpeech 2 paper are different from our settings: we use LibriTTS for both training and evaluation, while NaturalSpeech 2 uses Multilingual LibriSpeech (MLS) for training and LibriSpeech & VCTK for evaluation. Moreover, NaturalSpeech 2 did not open source its code. Therefore, we use [an open-source version of NaturalSpeech 2](https://huggingface.co/amphion/naturalspeech2_libritts), which is also trained solely on LibriTTS. The performance in the original paper cannot be directly taken for comparison.
> 2. UniAudio. UniAudio did open source its code so we can use its official checkpoint for inference. However, the evaluation code (the code to calculate WER and speaker SIM) is not open sourced. We tried to reproduce the results of ground truth data but failed (maybe due to the difference in text normalization). Besides, the evaluation data is also different -- UniAudio: LibriSpeech, UniFlow-Audio: LibriTTS (This will not favor UniFlow-Audio since LibriTTS is also a part of the training data of UniAudio: LibriLight + LibriTTS + MLS). Therefore we decided to do the inference of UniAudio ourselves, on the same test set: LibriTTS test-clean. The calculation of WER and SIM uses the same code as NaturalSpeech 2 and UniFlow-Audio. The code is included in the supplementary material `uniflow_audio/evaluation/tts.py`. Based on these differences, results in the original paper cannot be directly used while our reported results of different models are comparable and reproducible.
>
> Regarding the reason for the performance degradation, we also take a closer look at NaturalSpeech 2 and UniAudio's inference outputs:
> * We find that high WER of UniAudio mainly stems from cases where UniAudio fails to synthesize the given transcription correctly: appending redundant words or omitting words. This phenomenon is also observed for AR TTS models like VALL-E. The unsuccessful synthesis also negatively influences UniAudio's speaker SIM metric.
> * For NaturalSpeech 2, the model trained on LibriTTS exhibits limited audio quality, leading to ASR errors and high WER.
>
> The top-5 WER samples can be found in the revised supplementary `uniflow_audio/tts_baseline_high_wer_samples`. They are mostly short transcriptions.

---

> > ### Comment · Reviewer_DJhv · 2025-11-26
> > **Comments about the clarification**
> >
> > I appreciate the authors’ efforts in providing additional clarification. While the explanations are helpful, I would like to raise a few concerns regarding the comparative setup:
> >
> > - The reported performance for naturalspeech2 is substantially worse than that in the original paper, and the perceptual quality of the synthesized audio is also clearly inferior to the official samples. While I understand the intention to ensure experimental consistency, this discrepancy raises concerns, either the model requires substantial scaling to reach its intended quality, or the open-source implementation used here does not faithfully reproduce the official version. In such cases, it would be beneficial to mark these results explicitly as non-official reproductions, and, if possible, include additional baselines that are known to be more stable with different data size.
> > - As noted in the original paper, UniAudio uses sampling with post-selection during inference, a common approach for early (before 2025) autoregressive codec-based TTS systems. Since this design choice may confer certain advantages to the proposed method, providing a clear explanation of this inference pipeline within the main text would enhance transparency and fairness in comparison.
> > - Given the paper’s focus on flow-matching for generation tasks, and considering its relevance to related areas such as SVS and V2A, it would be more fitting to include flow-based TTS models (e.g., P-Flow, Glow-TTS, Flow-TTS) in the comparison. These baselines would more directly highlight the benefits of UniFlow’s ability to train across multiple datasets.
> >
> > Overall, I believe the authors present strong and promising work. A more careful treatment of the experimental setup,particularly in terms of baseline selection and reproduction clarity, would further strengthen the contribution.

---

> > > ### Author Response · Authors · 2025-12-01
> > > **Update of TTS evaluation setup**
> > >
> > > Thank you sincerely for your valuable suggestion! We have a closer look at the cross sentence setting of the test dataset we use (LibriTTS test). We find that for some utterances, the chosen speech prompt is too short or too long, which may underestimate the performance of baselines. Therefore, we change the TTS test set to the more standard test set, LibriSpeech-PC, and use the cross sentence setting in [F5-TTS](https://github.com/SWivid/F5-TTS/blob/main/data/librispeech_pc_test_clean_cross_sentence.lst), where prompt durations and punctuations are pre-processed. Besides, we use the source code of F5-TTS and re-train it on LibriTTS to obtain a stronger baseline. The model is trained for 200K steps to approximate the effective training samples used in training UniFlow-Audio. The updated TTS result is shown below. UniFlow-Audio demonstrates strong performance.
> > >
> > > Besides, we explicitly explain the fact that we do not apply any post-selection during inference of UniAudio, which leads to a much higher WER than the original paper.
> > >
> > >
> > > | Task | Model | Metrics | Results |
> > > |------|--------|-------------------|-------------------|
> > > | **TTS** | F5-TTS | WER $\downarrow$ \| SIM $\uparrow$ | 2.93 \| **58.0** |
> > > |         | UniFlow-Audio | | **2.19** \| 57.1 |
> > > |         | *UniAudio*    | | 11.93 \| 36.9 |

---

### Author Response · Authors · 2025-12-04
**Response Summary**

We thank all reviewers for their constructive feedback. We appreciate that all reviewers recognize the motivation of unifying TA and NTA audio generation tasks and our experimental results.
The main discussions and revisions are summarized as follows:
1. **Comparison with prior unified audio generation models (UniAudio, AudioX).**
All reviewers mentioned the comparison with previous unified generation models.
We provide detailed quantitative comparisons based on publicly available checkpoints. UniFlow-Audio achieves competitive or superior results with significantly less data and parameters. We explicitly note differences in training data and evaluation pipelines to ensure fairness and reproducibility. In terms of inference speed, UniFlow-Audio significantly outperforms UniAudio.
2. **Clarifying baseline reliability (especially TTS).**
Reviewer DJhv suggested reconsidering the baseline model of TTS.
We revised the TTS evaluation using the standard LibriSpeech-PC test set and stronger baselines (re-trained F5-TTS). Results demonstrate that UniFlow-Audio is competitive.
3. **Motivation for NAR unified modeling.**
Reviewer m9Bu has concerns about the motivation.
We refined our explanation: AR models rely solely on implicit alignment and suffer from exposure bias, making unified modeling brittle for long-sequence TA tasks. In addition, standard AR models predicts discrete tokens, which may be suboptimal for generation tasks. Our NAR design enables explicit alignment supervision, avoids exposure bias, and directly predicts continuous-valued audio representations.
4. **Architecture justification & ablations.**
We revise the manuscript to make the description of architecture design clearer and perform several additional ablations suggested by reviewers, e.g., validate the significance of explicit alignment, fusion depth analysis, the benefits of cross-task training.

Overall, the new empirical evidence and clarifications strongly support UniFlow-Audio as an effective and efficient unified NAR architecture for omnimodal audio generation. We believe the revisions and clarifications adequately address all concerns.

---

### Meta-Review · Area_Chair_rkQR · 2026-01-05

**Summary:**

The paper presents a method that can perform different audio generation task withing a single architecture. The paper explicitly divide tasks into time-aligned and non-time aligned and provide an architecture to handle both types.

The main concern of m9Bu was regarding novelty of the problem and the inherent alignment in the AR models vs NRA models. After the discussion authors response was that AR and NRA models provide complementary benefits, with AR models providing implicit alignment and NAR model avoid exposure bias and can benefit from explicit alignment. Area chair finds this discussion misleading, both model can enjoy implicit/explicit alignment. For example to make alignment explicit in AR models RoPE of the conditioning can be modified to align with generated signal. On the other hand to make alignment implicit in NAR models conditioning can be sequence concatenated to noisy tokens (the baseline which is not present in the paper). Thus area chair find this concern to be not addressed.

The main concern of the reviews DJhv and SnsA was regarding comparison with the baselines, mainly UniAudio and AudioX. During the discussion authors provide a lot of quantitative comparisons with these baselines. However the numbers seems to be questioned by DJhv. Moreover it seems that there is clear issues with presentation of qualitative results. For example comparisons from Tab 1, 2, 3 and UniAudio/ AudioX baselines are not present in the website. Given this, area chair believe this concern to be only partially addressed.

Overall area chair recommends a rejection for this paper, however encourage the authors to improve the presentation and separate the issue of alignment from auto-regression (potentially adapting the proposed framework for both AR and NAR models) and resubmit.

**Reviewer Concerns:**

See above.

**Reviewer Scores:**

m9Bu, DJhv, SnsA - stay the same

---

### Decision · Program_Chairs · 2026-01-26

Reject